# Learning High-Dimensional Perceptual Tasks with Hodgkin-Huxley Networks

## Abstract

This paper demonstrates that a computational neural network model using ion channel-based conductances to transmit information can solve standard computer vision datasets at near state-of-the-art performance. Although not fully biologically accurate, this model incorporates fundamental biophysical principles underlying the control of membrane potential and the processing of information by Ohmic ion channels. The key computational step employs Conductance-Weighted Averaging (CWA) in place of the traditional affine transformation, representing a fundamentally different computational principle. Importantly, CWA based networks are self-normalizing and range-limited. We also demonstrate for the first time that a network with excitatory and inhibitory neurons and nonnegative synapse strengths can successfully solve computer vision problems. Although CWA models do not yet surpass the current state-of-the-art in deep learning, the results are competitive on CIFAR-10. There remain avenues for improving these networks, *e.g.*, by more closely modeling ion channel function and connectivity patterns found in the brain.

## 1 Introduction

Deep learning has achieved groundbreaking advances in computer vision, but modern artificial neural networks have largely developed independent of neuroscience research on biological neuronal networks. Many of the core features found in biological neural networks have never been integrated successfully into artificial neural networks, raising serious questions of whether the success of deep learning is relevant to neuroscience research or *vice versa*.

This paper presents an artificial neuronal network incorporating features found in the classic Hodgkin-Huxley equations that relate neuronal membrane potential to the activity of ion channel conductance pathways. A primary feature of this approach is that the values that the neuronal elements of the network can take are range-limited and self-normalizing. In addition, each neuron is classified as either excitatory or inhibitory, an aspect of biological neural networks that until now has never been effectively utilized in artificial neural networks. Here we demonstrate that more biologically accurate networks can learn complex computer vision datasets such as CIFAR-10 competitively with the state-of-the-art just a few years prior. Biological neurons do not compute an affine transform of their inputs. Rather, the synaptic inputs compete against other active conductance pathways in the neuron for the control of the membrane potential and thus the firing state of the neuron. The valence of these different pathways is reflected by the zero-current potentials of the types of channel conductances being regulated, which determines whether the pathway excites or inhibits the neuron. The combination of these conductances into a single membrane potential is analogous to performing a conductance weighted averaging (CWA) of these zero-current potentials.

The concept of neuron valence is related to the types of synaptic channels that a neuron regulates in its post-synaptic neurons. Because neurons primarily release one type of neurotransmitter, they functionally fall into one of two classes: Excitatory or Inhibitory neurons.

In this paper, CWA similarly uses two valences, (+1) for neurons forming *excitatory* connections and (-1) for neurons forming *inhibitory* connections. Importantly, the combining of these pathways into a single neuron using CWA is constrained to produce neuronal outputs that lie somewhere between these two extremes, and thus the output of CWA is bound by the range of the valences.

Normalization and regularization strategies, such as BatchNorm (Ioffe & Szegedy, 2015) or DropOut (Srivastava et al., 2014), have been repeatedly shown to improve the quality of artificial neural networks. But CWA networks are by definition normalized and range-limited. Therefore, one can conjecture that CWA plays a normalization role in biological neural networks. As seen in Table **??**, bare CWA without additional normalization or regularization is competitive with standard affine networks that employ these strategies, especially in the case of fully connected networks (FCN), whose neurons, like biological neurons, are characterized by having thousands of inputs.

In sum, neural networks with CWA provide a more biologically accurate model with excitatory-inhibitory neurons and conductance-based normalization that succeeds on computer vision datasets without extra normalization and regularization, with implications for neuroscience and deep learning.

## 2 ABSTRACTING FROM HODGKIN-HUXLEY

Hodgkin and Huxley described the change in voltage within a neuron as a function of the ion channels active in the neuron (Hodgkin & Huxley, 1952), for which they received a Nobel prize. Their model remains the foundation of a modern understanding of biological neurons, but deep learning was never based directly on this model. In this section, a steady-state abstraction of the Hodgkin-Huxley model is developed, compared to standard (*i.e.*, affine) artificial neural networks, and used to derive the CWA neuron as an abstraction.

### 2.1 STEADY-STATE CONDUCTANCE MODEL

The information state of a neuron is encoded by the voltage on its surface membrane. The surface membrane is generally analogous to a capacitor; on its surface are many ion channels that transfer charge across this capacitance, allowing the movement of ions into or out of the cell. Depending on the ion species of a channel, the membrane potential will move in a direction determined by the equilibrium potential of that species. Ion flow changes the voltage across the membrane and is the key mechanism of information transfer within the brain. The strength of a synapse depends on the number of ion channels at the synapse, the ion species of the channel, and the overall activity of other pathways. Electrochemically, the net effect is a conductance-weighted average of the equilibrium potentials for all the ion channel pathways in the neuron (see *e.g.* Kandel et al. (2013), Ch. 5).

In the Hodgkin-Huxley model, the membrane potential (the voltage difference at the membrane) always moves in the direction that will reduce the net ionic current towards zero. The voltage and current are time-varying quantities governed by a differential equation, but they can be converted to a time-invariant, steady-state model by presuming that the the inputs to the network are constant. Because ion movement is linked to changes in the charge of the membrane capacitance, the steady-state membrane potential is found by letting the net ionic current go to zero.

For linear Ohmic conductance pathways, this stable membrane potential is determined by the conductance-weighted average of the zero-current potentials of the ionic pathways present in the neuron. For a single-chamber neuron $i$, and letting $j \sim i$ range over presynaptic neurons $j$, the voltage $V_i$ of $i$ can be written in simplified form as:

$$V_i = \frac{G_r E_r + \sum_{j \sim i} G_{i,j} p_j E_j}{G_r + \sum_{j \sim i} G_{i,j} p_j} \text{ where } p_j = \sigma \left( \frac{V_j - E_j^{\text{th}}}{k_j} \right) \tag{1}$$

This steady-state equation is the expected value of a probabilistic model driven by neural spike trains. The value $p_j$ represents the probability that the pathways leading out of neuron $j$ are open, that is, the synapses out of $j$ are releasing transmitter onto the post-synaptic ion channels. The exact computation of $p_j$ is the only element of Equation 1 not well defined by biophysics; in steady-state (non-spiking) models, it is commonly modeled as a logistic function or a ReLU. The logistic function in Equation 1 has a spike threshold $E_j^{\text{th}}$ and is scaled by a fixed scaling constant $k_j$. The $G_{i,j}$ terms describe the maximum conductance possible along the pathway from neuron $j$ to $i$, which occurs when $p_j = 1$. The term $G_r$ represents the resting self-conductance of the neuron, and $E_r$ the self-potential absent synaptic activity (see *e.g.* Gabbiani & Cox (2017), Chs. 2,11; Koch & Segev (1999), Chs. 1,2). These interactions are shown in Figure 1.

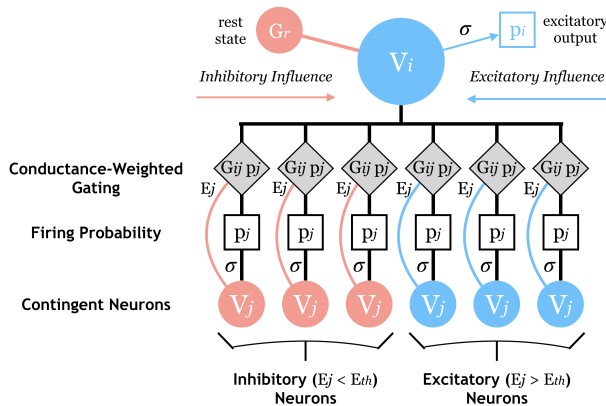

Figure 1: The steady state approximation of a biological neuron $i$ with excitatory valence. The neuron has its voltage $V_i$ determined as in Equation 1 by the firing probabilities $p_j$ of the contingent neuron, each of which has an influence gated by the conductance $G_{ij}$ of the synapse $ij$ and the firing probability $p_j$. This influence is either inhibitory (negative) or excitatory (positive) depending on the inherent, fixed state of the contingent neuron. There is also a resting inhibitory bias.

The $E_j$ terms are the zero-current potentials for each pathway, specific to the type of neuron $j$. For example, pyramidal neurons produce excitatory glutamatergic synapses yielding larger $E_j$ values, but lateral inhibitory interneurons produce GABA-ergic synapses with $E_j$ values below the threshold.

At behaviorally relevant time scales, changes in $p_j$ are solely responsible for neuronal membrane potential dynamics. Key factors causing changes in $p_j$ include changes in synaptic activity for neurotransmitter-gated channels or changes in gating of the other membrane channels due to voltage or intracellular messengers. For neurons, the importance of these changes in membrane potential are that they regulate neuronal spiking and hence neurotransmitter release onto follower neurons.

At larger time scales, the $G_{i,j}$ adapt to control how the neuron processes information based on the addition or removal of channels; the method of this adaptation is as yet not well understood.

All of the terms in Equation 1 have fixed sign: $G_{i,j}$, $p_j$, $E_j$, $G_r$, $E_r$, $E_j^{\text{th}}$, and $k_j$. In this model, a synapse is inhibitory if the incoming zero-current potential $E_j$ is less than the outgoing threshold $E_i^{\text{th}}$, in which case $j$ has an influence on $i$ that tends to push the spike probability below $\frac{1}{2}$. Thus neuron types with $E_j$ less than the threshold are described as *inhibitory*, and those with $E_j$ above the threshold are *excitatory*. Biological neural networks have several types of inhibitory and excitatory neurons, but *every* outgoing synapse of an excitatory neuron has an excitatory effect on the post-synaptic neuron, and conversely *every* outgoing synapse of an inhibitory neuron has an inhibitory effect. This partitions the network into classes of neurons with valences $E_j$ determined by the class.

## 2.2 COMPARISON TO AFFINE NEURAL NETWORKS

Artificial neural networks were originally modeled on biological neurons (Rosenblatt, 1958) but with simplifications. In particular, they assume that the total conductance is fixed (Shriki et al., 2003). In analogy to Equation 1, the typical artificial neuron is described by

$$z_i = b_i + \sum_{j \sim i} W_{i,j} p_j \quad \text{where} \quad p_j = \sigma(a_j), \tag{2}$$

where $\sigma$ is an activation function, typically a sigmoid such as the logistic or $\tanh$. Because $z_i$ is an affine transformation of the inputs $p_j$, such networks will be referred to as *affine networks* below.

Due to the long history of affine networks within machine learning, their connection with neuroscience is now obsolete, with advances in neuroscience having relatively impact on the machine learning community. The disconnection is further enhanced by the success of deep learning in computer vision and natural language processing, in comparison to neuroscience simulations which typically just seek

to mimic some aspect of experimental observations. One of the goals of this paper is to demonstrate experimentally that networks of CWA neurons are potentially as powerful as affine neural networks.

Equation 2 has similarities to Equation 1, but two aspects of Hodgkin-Huxley model are missing:

1. A conductance-weighted average (CWA) controlling neuronal activity.
2. A zero-current potential making neurons either excitatory or inhibitory.

With regard to (1), conductance-weighted averaging has a normalizing influence, limiting voltage to a range determined by the zero-current potentials $E_j$. Numerous methods in deep learning impose normalization on neural activation, but this biological mechanism has not been extensively explored.

The nature of the CWA operation is fundamentally distinct from the operation of an affine net. In Equation 1, the output $p_j$ of pre-synaptic neurons is *only* used in order to determine the conductance weighting, and the post-synaptic voltage is the weighted average of the incoming zero-current potentials $E_j$, which divides the neurons into excitatory and inhibitory classes, as in point (2) above.

Previous attempts to implement Equation 1 have been limited in scope and results. We demonstrate that an abstraction incorporating points (1) and (2) above can solve computer vision problems to a level that exceeds the state-of-the-art in affine networks from just a few years prior.

## 2.3 CONDUCTANCE-WEIGHTED AVERAGING

From Equation 1, the Hodgkin-Huxley neuron computes a weighted sum of the *fixed $E_j$* values, *i.e.*,

$$V_i = \sum_{j \sim i} w_{i,j} E_j \quad \text{where} \quad w_{i,j} = \frac{G_{i,j} p_j}{\sum_{k \sim i} G_{i,k} p_k}$$

Here $w_{i,j}$ reflects both *inputs* and *parameters*. For an input vector $p$ this equation computes a probability distribution $\mu(p)$ and takes the expected value over a vector of fixed values, labeled $e$:

$$z_i = \mathbb{E}_{\mu(p)}[e] = \sum_{j \sim i} e_j \, \mu(p)_j = \sum_{j \sim i} e_j \left\{ \frac{h(g_{ij}) \, p_j}{\sum_{k \sim i} h(g_{ik}) \, p_k} \right\} \tag{3}$$

The parameters $g_{ij} \in \mathbb{R}$ vary across the real line, and the function $h : \mathbb{R} \to [0, \infty)$ enforces nonnegativity. Equation 1 is obtained by setting $G_{i,j} = h(g_{ij})$ and supposing a connection to a bias node $r$ with $h(g_{ir}) = G_r$ with $p_r = 1$. Given that $p_j \in [0, 1]$, it follows that $z_i$ is necessarily range-limited, falling between the minimum and maximum values of $e_j$. For simplicity, set

$$e_j = 2 \frac{E_j - E_{\min}}{E_{\max} - E_{\min}} - 1$$

so that $e_j$ and $z_i$ fall in the interval $[-1, 1]$. Equation 3 is *Conductance-Weighted Averaging* or CWA.

Notice that Equation 3 is self-normalizing. The output $z_i$ necessarily lies in the range spanned by the $e_j$, so if $e_j \in [-1, 1]$, then $z_i \in [-1, 1]$. The factor $\mu(p)_j$ imposes normalization. The dependence on $p_j$ determines which of the $e_j$ to listen to, and the output is a weighted average of these values.

Affine neural networks alternate between affine transformation and non-linear activation. Equation 3 is already a non-linear transformation due to the denominator, and one might wonder whether non-linear activation is then necessary. In preliminary experiments, omitting a further activation function was sufficient to solve the MNIST digits problem with $98\%$ accuracy with a fully connected architecture, but obtaining good performance with more complex architectures and datasets did in fact require non-linear activation to silence minimally activated neurons, providing the benefit of sparse neural activation, known to contribute to good generalization in affine networks.

Rather than implementing $p_i$ as a sigmoid based on $E_i^{\text{th}}$ and $k_j$, we observe that the biophysics is not specific here and employ a simpler scaling and clipping function:

$$p_i = (z_i)_+ = \begin{cases} 0 & \text{if } z_i \leq 0 \\ z_i & \text{otherwise} \end{cases} \tag{4}$$

This equation is justified by the fact that as an expected value $z_i$ is bounded by the fixed values of the vector $e$ so that $z_i \in [-1, 1]$. Clipping off the negative values yields $p_i \in [0, 1]$ as desired;

it also results in ReLU activation (Nair & Hinton, 2010). The biological appropriateness of this simplification would depend on (1) the extent to which the spiking probability of biological neurons approximates a clipped linear function of membrane potential, and (2) whether there is functional significance to the balance of positive and negative $e_j$ with respect to the spike threshold in biological systems. These questions will not be resolved here, but a different biological activation function would likely not substantially change the overall computations resulting from Equations 3 and 4.

Now suppose in the extreme that rather than choosing $e_j$ anywhere in the interval $[-1, 1]$, one enforces either $e_j = +1$ or $e_j = -1$. If $e_j = +1$ neuron $j$ is *excitatory* or *positive*, and if $e_j = -1$, neuron $j$ is *inhibitory* or *negative*. This restriction loses no mathematical power of the system. As long as $\mu(p)$ can represent (nearly) arbitrary probability distributions, then $z_i$ can result in *any* point in the interval $[-1, 1]$ using just $e_j \in \{-1, +1\}$. Of course, in biological neural networks, there are several distinct values for $e_j$, not just two. It is not clear yet whether this distinction is computationally significant or merely a result of biological accident. Nonetheless, in implementing CWA, the experiments below divide the neurons up into two groups of positive/excitatory and negative/inhibitory neurons.

To explain these networks, Equation 3 can be rewritten as

$$z_i = \frac{1}{N_i(p)} \left\{ \sum_{j \text{ positive}} p_j \, h\left(g_{ij}\right) - \sum_{j \text{ negative}} p_j \, h\left(g_{ij}\right) \right\} \tag{5}$$

where $N_i(p) = \sum_j p_j \, h\left(g_{ij}\right)$ is the normalizing value. From an intuitive perspective, the left term as sums up the positive evidence and the right term sums the negative evidence. If the negative evidence predominates, then $z_i$ is negative. If the positive evidence predominates, then $z_i$ is positive. The parameters $g_{ij}$ control *which* evidence is listened to by neuron $i$.

Once $z_i$ is thought of as computing evidence for *something*, the role of neuron $i$ within a larger network is to represent that something as a discrete feature. If $z_i > 0$, the feature is observed to a degree determined by the magnitude of $z_i$. If $z_i \leq 0$, then on balance the feature has not been observed.

To summarize, CWA is an abstraction based on the Hodgkin-Huxley model. Its purpose is to demonstrate that networks using conductance-weighted averaging and partitioned into excitatory and inhibitory neurons can solve complex computer vision datasets. This is a more realistic model of biological neural networks than the affine model, but abstracted from reality in the following ways:

1) Non-synaptic channels (*i.e.*, $G_r$) are not dynamically gating and so their $p$ values are unversally 1. They behave similarly to the bias term in affine networks but also provide a mathematical anchor point for the normalizing sum.

2) Values for the zero-current potentials are scaled to lie in the range of $\pm 1$, and the channel conductances $G_{i,j}$ are represented by real-valued proxies $g_{ij}$.

3) Synaptic channel activity probability for a neuron is controlled by the membrane potential of the presynaptic neuron, and is modeled by a typical computational neuronal network activity function as in Equation 4 acting on the presynaptic neuron's membrane potential.

4) Only two types of zero-current potentials are used, $+1$ for excitatory neurons and $-1$ for inhibitory neurons.

The next sections discuss related work and the provide experimental results for CWA on computer vision tasks using classic fully-connected and convolutional architectures.

## 3 RELATED WORK

**Spiking Neural Networks.** Spiking neural network (SNN) models simulating the membrane potential as a differential equation have been applied to classification problems Pfeiffer & Pfeil (2018). The best supervised test accuracy for SNNs on MNIST is 99.31% with a convolutional SNN trained using backpropagation Lee et al. (2016). Several methods for SNNs achieve 98-99% on MNIST (Kulkarni & Rajendran, 2018; Diehl et al., 2015; Kheradpisheh et al., 2018). SNNs have also been run on CIFAR-10 with test accuracy of up to 92.37% (Hu et al., 2018; Sengupta et al., 2018). However, although SNNs compute with biologically realistic spikes, none of these models apply conductance

averaging as is present in biological neurons, and they rely on BatchNorm-inspired layers. Huh & Sejnowski (2018) published a method for supervised training in SNNs with a conductance-based rule, but they did not test the model on computer vision datasets. Diehl & Cook (2015) simulated SNNs with conductances and excitatory-inhibitory neurons on MNIST, reaching 95% accuracy on MNIST with an unsupervised technique. Diehl et al. (2015) used spiking affine networks, but with a renormalization scheme that divided all weights in a layer by the maximum positive activity to prevent unbounded spiking. This method is loosely reminiscent of but distinct from CWA. It achieved 99.15% test accuracy on MNIST in a convolutional SNN, similar to the 99.02% reported below.

**Excitatory-Inhibitory Networks.** Few classification results have been reported using neurons divided into excitatory and inhibitory groups. Diehl & Cook (2015) has already been mentioned. Delahunt & Kutz (2018) modeled the olfactory bulb of moths and applied it to MNIST with 75-85% accuracy. Zhu et al. (2017) introduced a method that modeled excitatory neurons as ReLU activated and inhibitory neurons as activated with a negative-valued saturating ReLU, attaining 94.22% on CIFAR-10 with networks containing both excitatory and inhibitory activations inside of an otherwise standard CNN; this result slightly exceeds that obtained with their model for excitatory-only networks, suggesting that there is value in using multiple neuron types.

**Normalization and Regularization Methods.** CWA keeps neural activity within a fixed range independent of the synapse strengths. Within deep learning, it is well known that large weights and activations can be a symptom of overfitting the training data. Techniques to control this include L2-regularization (Ng, 2004) of weights (*i.e.*, weight decay) and activities. An older technique, the Neocognitron, explicitly normalizes neural activities, inspired by the concept of complex vs. simple cells in the brain Fukushima et al. (1988). Dropout Srivastava et al. (2014) was proposed as a method of balancing the dependence of a neuron over its inputs, which also helps to prevent large variations in weight and activation magnitude. Krizhevsky et al. (2012) introduced Local Response Normalization as a method of locally smoothing neural activity in the hidden layers of a CNN. It was superseded by BatchNorm (Ioffe & Szegedy, 2015), which explicitly centers and rescales neural activities to a standard Gaussian. Most state-of-the-art results in deep learning rely heavily on BatchNorm to keep neural activities well-conditioned and to improve the curvature of the gradients during learning.

One hypothesis regarding conductance networks is that by imposing a finite range on the neural activities, CWA may play a role in the brain similar to that of BatchNorm within artificial neural networks. An advantage of CWA over BatchNorm is that it does not require extra parameters and does not impose requirements on how the training data are sampled in order to train the network. This last point is relevant since BatchNorm makes importance sampling difficult, which has prevented its use, for example, within deep reinforcement learning, where priority sampling is commonly applied. CWA could be used to fill the role of BatchNorm in these cases; as best as is currently known, the human brain is trained via reinforcement learning.

## 4 EXPERIMENTAL RESULTS

In order to assess its experimental value, CWA can be used as a drop-in replacement for the classical affine transformation inside of existing fully-connected and convolutional architectures. The resulting artificial neural networks and some variants have been tested on two image classification tasks, MNIST and CIFAR-10.The architectures, datasets, and results are described in this section.

### 4.1 COMPUTER VISION DATASETS

MNIST (LeCun & Cortes, 2010) consists of a training set of 60,000 images of handwritten digits from 0 to 9, with a test set of 10,000 images. All images are 28 x 28 pixels in size, and the dataset is balanced among the ten digits, so that each digit has 6,000 labeled training examples and 1,000 labeled test examples.

CIFAR-10 (Krizhevsky et al., 2009) contains 50,000 images for training and 10,000 for test, where the images are divided into 10 balanced classes such as *airplanes*, *cars*, *trucks*, and *dogs*. Each image in CIFAR10 has 32 x 32 x 3 pixels.

In the experiments below, the only data augmentation applied to CIFAR-10 was random shifting by up to 4 pixels followed by cropping, and random horizontal flipping of the image. This augmentation regime is sufficient to train common architectures to over 90% accuracy.

**Preprocessing.** Ordinarily, images are preprocessed before presentation to a neural network. This processing typically includes some form of whitening or contrast normalization, which makes the images more consistent and easier to classify. In the experiments below, only minimal preprocessing was performed. For CWA, the color channel values were scaled to lie between $[-0.3, 1]$ and then clipped to $[0, 1]$. Much like in Equation 4, this clipping was done to promote sparsity and had a positive effect on learning speed and generalization accuracy. For affine networks, the color channels were mean-centered and variance-scaled. This whitening could not be used for CWA because CWA assumes that the inputs lie in the interval $[0, 1]$.

Although these datasets are not sufficient to establish state-of-the-art performance for CWA, they do comprise a strong bank of tests to establish that CWA can solve computer vision tasks at the level of the state-of-the-art for affine networks from just 2-3 years prior.

## 4.2 ARCHITECTURES

One of the key advances in neural networks in the past decade has been the development of novel architectures, including the layout of artificial neurons and the introduction of components to address information-processing concerns. For CWA, three common architectures have been adapted for experiments on the datasets.

**Fully Connected Network (FCN).** FCNs consist of several layers of neurons with feed-forward computation. For CWA, the experiments partition each layer into excitatory and inhibitory neurons, with half the neurons in the layer being assigned to each category. The neurons in the input layer were doubled, with one excitatory copy and one inhibitory.

**Convolutional Neural Network (CNN).** Convolutional neural networks adapt a feed-forward network to incorporate local connections (receptive fields) and shared weights. The experiments below use a sequence of four or five convolutional layers with stride 2 and no pooling. To implement CWA-CNNs, the simplest adaptation is to assign each presynaptic channel a valence (*i.e.*, zero-current potential) in the context of the receptive field. For the experiments, half of the channels were considered excitatory and half inhibitory. To adapt the input image, the input channels were doubled from three to six, with two channels for each color, one excitatory and one inhibitory. Using tensor notation, for the standard convolutional filter one writes

$$\hat{z} = b + W \star p,$$

which is adapted to the conductance network as

$$\hat{z} = \frac{G_r e_r + (G \otimes e) \star p}{G_r + G \star p} \tag{6}$$

where $G_{ijxy} = h(g_{ijxy})$ is the max conductance parameter, $\otimes$ indicates componentwise multiplication (also known as the Hadamard product) and

$$e_{ijxy} = \begin{cases} +1 & \text{if } j < J/2 \\ -1 & \text{otherwise,} \end{cases}$$

where $J$ is the number of presynaptic channels to layer $i$, *i.e.*, half the channels are excitatory and half inhibitory. Here $G \otimes e$ is modeled rather than $p \otimes e$ to reflect the concept that the neuron valence is determined by the receptive field, but the two concepts are of equivalent significance given the definition of $e$ above.

**Residual Networks (ResNet).** Residual Networks (He et al., 2016) are a variant of convolutional neural networks characterized by blocks of successive convolutions with shortcut connections. This paper uses the ResNet-18 architecture without BatchNorm but with CWA convolutions as in Equation 6 and activated according to Equation 4.

## 4.3 TRAINING RESULTS

Each of the three architectures was tested on both datasets for both affine networks and CWA. For clarity, the affine networks used ReLU activation (Nair & Hinton, 2010) and were initialized with

| Architecture | MNIST | | CIFAR-10 | | | |
|---|---|---|---|---|---|---|
| | CWA | AFFINE + BN | CWA | CWA+BN | AFFINE NO BN | AFFINE + BN |
| FCN | 98.65 | 98.72 | 61.98 | 64.93 | 65.10 | 63.93 |
| CNN | 98.89 | 99.42 | 78.09 | . | 73.40 | 88.60 |
| RESNET-18 | 99.02 | 99.59 | 85.06 | . | 79.60 | 91.25 |

Table 1: Test Accuracy for CWA and Affine Networks on Two Datasets.

Glorot initialization (Glorot & Bengio, 2010). Two variants of affine networks were tested. The first variant (Affine, No BN) took the same unwhitened, scaled inputs as CWA and did not use batch norm. In the second variant (Affine + BN), image inputs were whitened and each ReLU activation was preceded by an application of BatchNorm. The purpose of training affine networks is to obtain meaningful baselines for comparison. In one case, CWA was tested with BatchNorm applied similarly (CWA+BN) The training procedures are described below, followed by the results.

**Initialization.** The $g_{ij}$'s and $g_{ijxy}$'s comprise all of the parameters in a conductance network. These parameters were Glorot-initialized, so that the initial $G_{i,j}$ parameters varied tightly around 1. Unlike with standard neural networks, CWA parameters can be initialized to zero without preventing learning, although we did not do so. To enforce the nonnegativity of $G_{i,j}$, $h(g_{ij}) = (1 + g_{ij})_+$ was used in Equation 3. Although this function is discontinuous, by initializing the $g_{ij}$ close to zero, a wide range is available for optimization, and the linear scaling of the gradient proved to be advantageous from a numerical perspective. By contrast, $h = \exp$ could have been used, but it caused overflow at later stages of training in some cases during preliminary experiments.

**Bias Term.** The bias conductance $G_r = h(g_r)$ and its valence $e_r$, representing the neuron's own zero-current potential, were fixed and not adapted during training. Specifically, $G_r = 1$ was used, and $e_r = -0.4$ was used for FCN but $e_r = 0$ for CNN and ResNet. By fixing these values, the denominator in Equations 3 and 6 could never produce division by zero. The inhibitory bias for FCN was determined experimentally in order to produce a tendency towards not firing in the zero-input state. In convolutional architectures, setting $e_r$ to any value other than zero caused training to fail in preliminary experiments for unknown reasons.

**Categorical Output.** The output of an affine network on categorical networks is traditionally determined by applying a softmax activation, $\mathbb{P}(c_i) \propto \exp(z_i)$, in place of the standard activation. Biological networks do not seem to have categorical outputs of this form. Thus in order to compute a categorical output, a final affine layer with softmax activation was added as the top layer of each CWA architecture to extract the categories from the CWA network.

**Parameter Fitting.** All networks, both CWA and affine networks, were trained with backprop using mini-batch stochastic gradient descent with the Adam optimizer (Kingma & Ba, 2014) to control per-parameter learning rates ($\beta_1 = 0.9$, $\beta_2 = 0.99$). Lower initial learning rates ($\eta = 1e-5$) were found to be necessary in most cases for CWA, and the resulting parameter learning was noticeably slower than for affine networks, which were trained with initial $\eta = 0.001$. Backprop was used purely as a mathematical method of demonstrating that CWA networks do indeed have the capacity to fit the data.

**Training Procedures.** Each network was trained until it had fit the training set to high accuracy. In all cases the training accuracy was $> 99.4\%$. At each epoch of training, every training image was presented to the network in minibatches. At the end of each epoch, the training loss was computed, and if 25 epochs elapsed with no improvement in training loss for CWA, the learning rate was decreased by a factor of 10. For affine networks, only 5 epochs without decrease were allowed. The test accuracy was computed after each epoch, and the lowest test accuracy during training is reported for each network. In general, test accuracy did not substantially decrease at any point during training. No validation sets were used; all networks were trained on all training data.

**Network Sizes.** The weight-bearing layers were of the same size for both CWA and affine networks in all cases, but varied according to dataset. For MNIST, the FCN had two hidden layers of $2,000$ neurons each. For CIFAR-10, a five-layer FCN with three hidden layers of $2000, 1000, 500$ was used. The CNN for MNIST had four convolutional layers with 32, 64, 128, and 256 channels, respectively. The CNN for CIFAR-10 added a fifth convolutional layer with 512 channels. ResNet-18 consists of four blocks of two convolutions each. The initial input was projected to 64 channels, and the four

blocks internally used 64, 128, 256, and 512 channels, respectively. The ResNet-18 architecture was not changed across tasks.

**Results.** The results are shown in Table **??**. Focusing first on the fully connected architecture, CWA is seen to be competitive with affine networks with BatchNorm. On MNIST, the two are nearly equal at 98.65% and 98.72%, and on CIFAR-10 they differ by less than 2% at 61.98% and 63.93. Adding BatchNorm to CWA does however improve performance to 64.93%. There are many opportunities to close the gap between CWA and Affine networks. The input and output procedures are not biologically driven and may not be optimal for CWA. For example, the affine networks have the benefit of using whitened inputs, whereas the CWA inputs are merely scaled and clipped. Different connectivity patterns between excitatory and inhibitory inputs may also help as well.

In the case of the CNN and ResNet-18, there is a greater distance between the results for CWA and affine networks, but the results for CWA on CIFAR-10 are still comparable to published results from approximately 2012 to 2014 (see *e.g.* `http://rodrigob.github.io/are_we_there_yet/build/`). It is also noteworthy, that without BatchNorm, affine networks underperform CWA. One reason why the convolutional version of CWA performs worse might have to do with the low number of incoming connections. The first convolutional kernel in CNN has merely 6 x 5 x 5 = 150 connections for each neuron in the second layer. This number should be compared with 32 x 32 x 6 = 6,144 connections for each neuron in the second layer of the FCN and tens of thousands of connections neurons in the brain. Higher connectivity may be important to good performance with CWA, because it allows the denominator of Equation 3 to depend on many more input features. When there are fewer connections, a handful of input features can dominate the denominator, reducing generalization performance. Wider convolutional kernels might help, but are impractical on small images such as those of CIFAR and MNIST. Better performance might be possible on datasets of larger images. It must be added that as an architecture, CNNs uses weight sharing in order to achieve economy with respect to data and memory. Biological neural networks are organized differently, and it may be that a new architectural paradigm is needed for CWA.

Regardless of any deficiencies, the results above demonstrate that a network with conductance averaging and excitatory-inhibitory neurons such as CWA can solve real problems. To our knowledge, this is the first time that these two biological features have been successfully combined inside of an artificial neural network.

## 4.4 CWA Variants: AbsNorm and Lateral Inhibition

Networks with excitatory and inhibitory neurons and strictly positive weights are a consequence of Equation 1, but there is a generalization of conductance averaging that can be applied to affine nets as an ablation study on CWA. The basic idea is to make the valence $e_j = \pm 1$ a property of the synapse ($e_{i,j}$) rather than the neuron and then assign $W_{i,j} = G_{i,j} e_{i,j}$ so that $|W_{i,j}| = G_{i,j}$. Then Equation 3 becomes

$$z_i = \sum_{j \sim i} \frac{W_{i,j} \, p_j}{\sum_{k \sim i} |W_{i,j}| \, p_k},$$

which is just Equation 2 with divisive norming based on the absolute value of the weights. This equation will be called AbsNorm. As with CWA, it forces $z_i \in [-1, 1]$, so that applying a ReLU afterwards results in $p_i \in [0, 1]$.

AbsNorm characterizes the two main differences between CWA and affine nets, namely, the divisive averaging based on total input activity and the pattern in which outgoing connections from a neuron all share the same valence. Interestingly, AbsNorm failed to perform as well as CWA in preliminary experiments with the FCN architecture on CIFAR-10; these results are shown in Table **??**. The difference between CWA and AbsNorm is substantial, and was consistently worth about 2% test accuracy in four separate trials, the highest of which is shown for each in the table. This result suggests that there is value in separating the neurons into distinct valence classes. Distinct valences delineate of features as having an inherently positive or negative significance, though it is not clear why this yields better performance at this time.

One major area in which the architectures tested in this paper are not biologically faithful is the connectivity pattern among different neuron types. This is an interesting aspect for future work. As a first step in this direction, extra inhibitory neurons were added to each layer of CWA and

| Architecture | Test Accuracy |
|---|---|
| CWA | 61.98 |
| AbsNorm | 59.95 |
| CWA + Lateral Inhibition | 62.03 |

Table 2: CWA Variants, FCN Accuracy on CIFAR-10

activated recurrently in a temporal rollout, simulating the effect of lateral inhibitory interneurons, which potentially regulate the overall activity level within cortical regions. As seen in Table **??**, CWA with this addition did achieve the highest accuracy among CWA variants tested, but only marginally so. More work is needed to establish the effect of different connectivity patterns.

## 5 Conclusions

**Limitations & Future Work.** There are several ways to improve CWA. Most prominently, the decision to split layers into half excitatory and half inhibitory neurons connected in a feed-forward fashion was somewhat arbitrary and does not reflect biology. It would be interesting to approximate biological valence ratios and connectivity patterns to understand better how these features contribute to information processing in neural networks.

Realistic connectivity patterns would require the use of a recurrent activation pattern rather than feed-forward. Although substantial work has been done simulating the dynamics of spiking neural networks, less research has been directed towards discrete time slice approximations as used in recurrent neural networks (RNNs). The same principles of this paper could be applied to that case as well.

Separately, it is notable that the fully connected architecture obtained closer results to the affine baseline than convolutional architectures. Convolutional neurons in deep learning generally have fewer incoming connections than biological neurons do, and in may be that high connectivity is needed for CWA to work well. This issue needs further exploration.

Finally, parameter fitting with CWA was based on backprop. There is no conclusive evidence that animal brains adapt their conductances using backprop, and it would be worthwhile to explore alternatives for parameter fitting that do not use backprop but work well with CWA.

**Applications to Deep Learning.** CWA inherently implements features that have been found to be useful in deep learning, such as sparsity and limiting the output range of neurons. CWA does so by dynamically scaling the output to marginalize out the level of input activity. It would be of interest to understand how this dynamic scaling affects other properties of affine networks, such as susceptibility to adversarial images. Furthermore, the network architectures tested in this paper as well as the training procedures were developed for affine networks. It may be that with specially adapted training procedures or architectures, CWA could still achieve state-of-the-art results.

**Applications to Neuroscience.** In contrast to many biologically motivated neuronal simulations, the CWA experiments in this paper solve serious computer vision problems, suggesting that the value of neuroscience research can be illustrated by application to deep learning problems. However, non-biological adaptations were required to process the inputs and outputs of the network and to control parameter adaptation. It would be interesting to understand the extent to which more biologically accurate settings can still obtain similar results.

The experimental setting of this paper also offers a miniature lab in order to test questions such as whether multiple neuron valences promote better learning, or how the balance and connection among these valence types affect information processing and learning. Although such experiments could not conclude anything about what biological systems actually do, they could be invaluable in identifying possibilities for consideration.

To conclude, CWA implements a biologically accurate understanding of neuronal activity and for the first time achieves competitive performance at computer vision datasets in a network with conductance averaging and excitatory-inhibitory neurons. This research provides a pattern for identifying and abstracting basic biophysical knowledge to demonstrate relevance to machine learning.

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

## 6 SUPPLEMENTARY MATERIAL

### 6.1 CODE IMPLEMENTING CWA

The following code implements a fully connected CWA layer in python as a PyTorch module. As can be seen, CWA requires two matrix multiplies instead of the usual one.

```python
class CWALayer(torch.nn.Module):
    def __init__(self, neurons_in, neurons_out):
        super(CWALayer, self).__init__()
        self.neurons_in = neurons_in
        self.neurons_out = neurons_out
        self.gs = torch.nn.Parameter(
            torch.zeros(neurons_in, neurons_out)
        )
        torch.nn.init.xavier_normal_(self.gs)
        self.es = torch.nn.Parameter(
            torch.zeros(neurons_in).fill_(1.0),
            requires_grad=False
        )
        self.es[:neurons_in//2] = -1.0

    def fix(self, gs):
        return torch.clamp(1+gs, 0.0, float('inf'))

    def forward(self, ps):
        Gs = self.fix(self.gs)
        normalizer = 1 + torch.matmul(ps,Gs)
        numerator = -0.4 + torch.matmul(ps*self.es.unsqueeze(0), Gs)
        activity = (
            numerator / torch.clamp(normalizer, 1e-6, float('inf'))
        )
        return activity
```

The following code implements a convolutional CWA layer. As is seen below, implementing CWA requires two convolutions instead of one, but is otherwise similar to the affine version.

```python
class ConvCWALayer(torch.nn.Module):
    def __init__(self, in_channels, out_channels,
                 kernel_size, stride=1, padding=0):
        super(ConvCWALayer, self).__init__()
        self.gs = torch.nn.Parameter(
            torch.zeros(out_channels, in_channels,
                        kernel_size, kernel_size)
        )
        torch.nn.init.xavier_normal_(self.gs)
        self.es = torch.nn.Parameter(
            torch.zeros(in_channels, kernel_size,
                        kernel_size).fill_(1.0),
            requires_grad=False
        )
        self.es[:in_channels//2] = -1.0
        self.stride = stride
        self.padding = padding

        self.grest = torch.nn.Parameter(torch.zeros(1))

    def fix(self, gs):
        return torch.clamp(1+gs, 0.0, np.inf)
```

```
def forward(self, ps):
    Gs = self.fix(self.gs)
    normalizer = 1.0 + torch.nn.functional.conv2d(
        ps, Gs, stride=self.stride, padding=self.padding
    )
    numerator = torch.nn.functional.conv2d(
        ps, Gs*self.es.unsqueeze(0),
        stride=self.stride, padding=self.padding
    )

    return (
        numerator / torch.clamp(normalizer, 1e-6, float('inf'))
    )
```

