# OpenReview forum: "LEARNING DIFFICULT PERCEPTUAL TASKS WITH HODGKIN-HUXLEY NETWORKS"
_ICLR.cc/2020/Conference — Reject_

### Official Review · AnonReviewer3 · 2019-10-23
**Official Blind Review #3**

**Rating:** 6

**Review:**

This paper proposes a novel neural network architecture inspired by the analysis of a steady-state solution of the Hodgkin-Huxley model. Using a few simplifying assumptions, the authors use conventional backpropagation to train DNN- and  CNN-based models and demonstrate that their accuracies are not much lower than the state-of-the-art results.

The paper is well-written, sufficiently detailed and understandable. Derived self-normalizing Conductance-Weighted Averaging (CWA) mechanism is interesting in itself, especially contrasting CWA results with those obtained for the non-Batch-Normalized networks. It is also inspiring to see that this model can be derived based on a relatively accurate biological neuron model.

My main question is actually related to the potential impact of this study. I am curious about the implications and the ways in which these results can inspire other researchers.

After reading the paper, I got an impression that:

(a) From the point of view of a machine learning practitioner, these results may not be particularly impressive. They do hint at the importance of self-normalization though, which could potentially be interesting to explore further.

(b) From the point of view of a neuroscientist, the proposed model might be too simplistic. It is my understanding, that neural systems (even at "rest") are inherently non-equilibrium (and I assume the presence of simple feedback loops could also dramatically change the stead-state of the system). Is it possible that something similar to this "steady-state inference" mode could actually take place in real biological neural systems?

(c) Presented results appear to be important from the point of view of someone who wants to transfer insights from biology into the field of deep learning. But there might be an extent to what is achievable given a simple goal of optimizing a supervised accuracy of an artificial neural network trained using gradient descent (especially considering limitations imposed by hardware). I am optimistic about the prospect of knowledge transfer between these disciplines, but it is my feeling that the study of temporal dynamics, emergent spatiotemporal encodings, "training" process of a biological neural system, etc. have potentially much more to offer to the field of machine learning. These questions do appear to be incredibly complex though and the steady-state analysis is definitely a prerequisite.

**Experience Assessment:**

I have read many papers in this area.

**Review Assessment: Checking Correctness Of Derivations And Theory:**

I assessed the sensibility of the derivations and theory.

**Review Assessment: Checking Correctness Of Experiments:**

I assessed the sensibility of the experiments.

**Review Assessment: Thoroughness In Paper Reading:**

I read the paper at least twice and used my best judgement in assessing the paper.

---

> ### Author Response · Authors · 2019-11-15
> **Response to Review 3**
>
> Thank you for your encouraging comments. Regarding the implications of this work, the primary implication is that neuroscience models can be compared with deep learning results on deep learning benchmarks and with appropriate modelling, the results can be competitive, as one would expect given that the brain performs all these tasks. Furthermore, these comparisons raise questions for neuroscientists that can spur further research. For example, if batch normalization performs better than CWA, why does the brain use CWA? Is it due to evolutionary convenience, or are there other, unexplored benefits to CWA that can explain the choice?
>
> Below, we respond to the listed impressions in turn.
>
>
> Comment: (a) From the point of view of a machine learning practitioner, these results may not be particularly impressive. They do hint at the importance of self-normalization though, which could potentially be interesting to explore further.
>
> Response: We agree with this assessment; ML practitioners may find some ideas here, but are unlikely to use CWA as a primary method. There are cases, though, where batch norm is not practical or possible, such as in reinforcement learning, and it might be interesting to explore CWA as a normalization method in this context.
>
> Comment: (b) From the point of view of a neuroscientist, the proposed model might be too simplistic. It is my understanding, that neural systems (even at "rest") are inherently non-equilibrium (and I assume the presence of simple feedback loops could also dramatically change the stead-state of the system). Is it possible that something similar to this "steady-state inference" mode could actually take place in real biological neural systems?
>
> Response: CWA represents limit behavior of the Hodgkin-Huxley differential equations, so it may be possible. Given the need to use spiking to transmit information, the mammalian nervous system appears quite chaotic at the granular level.  However, it is equally true that your perceptions, a product of your brain, are often quite stable.  Spiking, however, is not an essential feature of neuronal function, as many sensory neurons, such as photoreceptors perform computations in the manner we are using here to adjust synaptic release without spiking.   In fact, time dependence is readily incorporated in our model by including a capacitive term.  In this case, the CWA determines the state the neuron is heading towards, and the time constant how quickly it will get there.  The current data sets we have been optimizing against do not require such temporal precision so this has not been incorporated yet into our model.
>
> Comment: (c) Presented results appear to be important from the point of view of someone who wants to transfer insights from biology into the field of deep learning. But there might be an extent to what is achievable given a simple goal of optimizing a supervised accuracy of an artificial neural network trained using gradient descent (especially considering limitations imposed by hardware). I am optimistic about the prospect of knowledge transfer between these disciplines, but it is my feeling that the study of temporal dynamics, emergent spatiotemporal encodings, "training" process of a biological neural system, etc. have potentially much more to offer to the field of machine learning. These questions do appear to be incredibly complex though and the steady-state analysis is definitely a prerequisite.
>
> Response: Yes, the last point seems to be key. A better understanding of the brain should lead to better machine learning, but the path to get there may not be direct. At present deep learning seems to outstrip neuroscience in terms of generating intelligent behavior, and the question is why? Answering this question requires putting neuroscience into a setting where it can be compared directly with machine learning, and that is why we have pursued the question of performance on benchmarks: it should raise questions for neuroscientists that help to address the gaps that exist at present. The identification and understanding of these gaps should lead to gains in the field of neuroscience, which could then benefit machine learning as well. The present study is a somewhat humble step in this direction, but it does offer some novel components, including as far as we are aware the first network with partitioning among excitatory and inhibitory neurons that performs acceptably on benchmark tasks. We are working on temporal dynamics, but there the modeling and computation become quite a bit more complex, which is why this present research is a necessary first step.

---

### Official Review · AnonReviewer2 · 2019-10-23
**Official Blind Review #2**

**Rating:** 3

**Review:**

This paper focuses on non-spiking Hudghkin-Huxley model, which is different from existing works on spiking neural network-based Hudghkin-Huxley model.

There are many ways of using neuron firing model as unit to construct neural networks. They choose a specific way (mentioned above). I think the most interesting part would be the CWA method which achieves the normalization.

They have a fair list of literature in spiking neural networks. But I find the way they illustrate the difference between their model and other models is insufficient. They should focus on the model-wise difference, instead of focusing on whether it’s applied to MNIST or not or what’s the accuracy.

They don’t include any other SNN model in the paper for experimental comparison. They also mention a few SNN works that work well on MNIST in the related work section which actually have better accuracies than their model. So it is inappropriate to say this proposed method is a state-of-art neuro-inspired method, Because others perform well on MNIST as well, and their limited experiments only investigate MNIST and CIFAR-10, which are less interesting generally.

CWA cannot outperform Affine+BN.

Overall, the idea is somehow interesting, but the experiments are weak. Applying the method to MNIST and CIFAR-10 is far from being called either “interesting computer vision applications” or “difficult perceptual tasks”. They only use perceptual in the title, but the applications are MNIST and CIFAR-10. It feels like they want to learn something big, but they only focus on benchmark datasets.

They compare nothing with other SNN type of model on other truly difficult perceptual tasks.


**Experience Assessment:**

I have read many papers in this area.

**Review Assessment: Checking Correctness Of Derivations And Theory:**

I carefully checked the derivations and theory.

**Review Assessment: Checking Correctness Of Experiments:**

I carefully checked the experiments.

**Review Assessment: Thoroughness In Paper Reading:**

I read the paper thoroughly.

---

> ### Author Response · Authors · 2019-11-15
> **Responses to Reviewer 2**
>
> Thank you for reviewing our paper. Our goal was not to study spiking models but to investigate a static-time realization of conductance averaging. We do consider the SNN results to be relevant and hence included several current links into the literature. SNN research brings one aspect of biological networks into artificial neural networks (spiking), while our research brings different aspects, namely, that neuron activity is naturally normalized and range limited by physical limits on the transmembrane movement of ions. We also show that modeling this aspect of neural behavior naturally leads to the introduction of inhibitory and excitatory neurons, which is an interesting recapitulation of real biological networks.
>
> In what follows, we provide responses to some specific comments.
>
> Comment: …it is inappropriate to say this proposed method is a state-of-art neuro-inspired method, ... CWA cannot outperform Affine+BN.
>
> Response: Our claim was not that CWA was state-of-the-art, but that it was competitive with the state-of-the-art. More importantly, our goal is actually to demonstrate something about the neuroscience rather than to introduce a new practical method based on neuroscience. Thus “neuro-inspired” was not what CWA was intended to accomplish.
>
> Many if not most truly state-of-the-art results in deep learning require substantial engineering in the training methods and network architecture. We have avoided this kind of engineering in order to perform a basic controlled comparison between CWA and affine networks. The results for affine+BN demonstrate how wide this gulf is; 64% is hardly a strong result on CIFAR-10 given what we know is possible. The point is not to demonstrate superiority for CWA but rather mere competence. We want to show that basing a network on Hodgkin-Huxley inspired principles doesn’t destroy performance -- something which has not been shown before.
>
>
>
> Comment: Applying the method to MNIST and CIFAR-10 is far from being called either “interesting computer vision applications” or “difficult perceptual tasks”. They only use perceptual in the title, but the applications are MNIST and CIFAR-10. It feels like they want to learn something big, but they only focus on benchmark datasets.
>
> Response: Object recognition is of course a perceptual task. We can see where our use of the word “difficult” in the title could set expectations in the wrong way; we have edited the title to replace the word “difficult” with “high-dimensional”, which will hopefully alleviate this issue. Please note that we cannot change the title in the system at this time, but the paper submitted has been revised.
>
> Once again, our goal was to demonstrate that conductance averaging as performed in biological neurons can succeed at the tasks we tested, which has not been shown previously and is accomplished by these experiments. Our main goal was to test how incorporation of biophysical properties, such as those used in real neurons, affects computations produced by artificial neurons. The use of benchmark datasets is standard in work on artificial neural networks, whence the selected experiments.  Whether this will lead to “something big” can only be determined over time, but we have demonstrated that neuronal biophysical properties can be readily incorporated into computational networks, which will hopefully lead to better modeling and analysis of real biological circuits using the tools that have been developed to create and study artificial neural networks.
>
> Comment: They compare nothing with other SNN type of model on other truly difficult perceptual tasks.
>
> Response: As a first point, CWA is not a competitor to SNN models; at present, we are not aware of results in spiking neural networks that incorporate conductance averaging, which does happen in biological neurons, other than highly complex simulations in specialized programs such as NEURON and GENESIS, which have not been easy to optimize or to use to solve high-dimensional perceptual tasks. If fact, we do see a clear path by which CWA could be combined with spiking networks, or used in neuromorphic chip design in future work. CWA as formulated in this paper is static and not temporal, as can be seen from the implementing code we have added to the supplementary materials.  However, the paths to add temporal features to the network is clear from the Hodgkin and Huxley model and will be a focus of our future work.
>
> We did not compare with SNNs because there is no simple, controlled comparison there. Biological neurons operate on spike trains, and they also perform conductance averaging. There simply is no conflict, and no comparison would be fair without also including spiking networks that incorporate conductance averaging, which we have not yet explored.
>
> There are certainly many opportunities to include spiking with this CWA work as future work, but the point here is to merely demonstrate that conductance averaging works competitively.

---

### Official Review · AnonReviewer1 · 2019-10-29
**Official Blind Review #1**

**Rating:** 3

**Review:**

I wanted to first thank the authors for their work on connecting biologically inspired CWA to deep learning. The paper overall reads nicely.

The authors make the assumption that reviewers are fully aware of biological terminology related to CWA, however, this may not be the case. It took a few hours of searching to be able to find definitions that could explain the paper better. I suggest the authors add these definitions explicitly to their paper (perhaps in supplementary), also an overview figure would go a long way. I enjoyed reading the paper, and before making a final decision (hence currently weak reject), I would like to know the answer to the following questions:

1. Have the authors considered the computational burden of Equation 3? In short, it seems that there are two summations (one for building the probability space over measure h) and one right before e_j. This is somewhat important, if this type of neural network is presented as a competitor to affine mapped activations.

2. It would be nice to have some proof regarding universal approximation capabilities of CWA. In my opinion it is, but a proof would be nice (however redundant or trivial - simply use supplementary).

3. I was a bit confused to see CWA+BN in the Table 1. In introduction, authors write “But CWA networks are by definition normalized and range-limited. Therefore, one can conjecture that CWA plays a normalization role in biological neural networks.” Therefore, I was expecting CWA+BN to work similarly as CWA for CIFAR10. Please elaborate further on this note.

4. Essentially, the CWA changes the definition of a layer in a neural net. Do authors see a path from “CWA works” to “CWA works better than affine?”. If so, please elaborate. Specifically, I am asking this question “Why should/must we stop using affine maps in favor of CWA?”. Now this may or may not be the claim of the paper. It’s ok if it is not; still showing competitive performance is somewhat acceptable, but certainly further insight would make the paper stronger.


**Experience Assessment:**

I have published in this field for several years.

**Review Assessment: Checking Correctness Of Derivations And Theory:**

I carefully checked the derivations and theory.

**Review Assessment: Checking Correctness Of Experiments:**

I carefully checked the experiments.

**Review Assessment: Thoroughness In Paper Reading:**

I read the paper thoroughly.

---

> ### Author Response · Authors · 2019-11-15
> **Responses to Reviewer 1**
>
> Thank you for these comments and questions. We would like to emphasize that the purpose of CWA is to demonstrate that biological concerns can be realized inside of neural networks that can be trained effectively, including channel conductance normalization and excitatory/inhibitory partitioning of neurons. Although we hope these insights are useful for deep learning practitioners as well, that is not the primary goal of this work. Answers to questions follow.
>
> Comment: I suggest the authors add these definitions explicitly to their paper (perhaps in supplementary), also an overview figure would go a long way.
>
> Response: A diagram showing the basic types of connections determining the cell voltage has been added as Figure 1 based on the steady-state biological model. Although we have not been able to do so during the comment period, a glossary can also be added to the supplementary material to clarify unfamiliar terms.
>
>
> Comment: Have the authors considered the computational burden of Equation 3? In short, it seems that there are two summations (one for building the probability space over measure h) and one right before e_j. This is somewhat important, if this type of neural network is presented as a competitor to affine mapped activations.
>
> Response: A PyTorch implementation of CWA layers has been added to the supplementary material. Our implementation of fully connected CWA has two matrix multiplies instead of the usual one matrix multiply for affine networks. Similarly, convolutional CWA has two convolutions instead of one. In our experiments, CWA is about 1.5x slower to run than affine networks overall. CWA is not proposed as a competitor to affine networks, but rather as an idealized model of the computation occuring in biological neurons, which may be useful both for neuroscientists and deep learning practitioners.
>
> Comment: It would be nice to have some proof regarding universal approximation capabilities of CWA.
>
> Response: At this time we do not have a proof of universal approximation capabilities, though we would be surprised if CWA did not have such capabilities, since it reflects the operation of the brain. The main reason for proposing CWA is that it reflects the way neurons in the brain use conductance pathways to perform computations.
>
> In terms of mathematical tools, it is generally true that every point within a convex set is the barycenter (expected value) of some probability distribution over its extreme points (the Choquet theorem). In this case, the e_j’s take on values in {-1,+1}^d where d is the size of the layer, and this includes all the extreme points of the convex set [-1,+1]^d representing the output voltages. Therefore all possible voltage outputs can be obtained. It remains to show that the inputs are sufficient to generate the necessary distributions in the course of one or more CWA layers. We leave this problem for a later time.
>
> Comment: I was a bit confused to see CWA+BN in the Table 1.
>
> Response: BN adds extra parameters to the network, which may explain why CWA+BN behaves better than CWA alone.  As a normalization technique, BN seems to outperform CWA. However, CWA has a biological implementation, which BN definitely does not. It should be noted in this context that CWA outperforms affine networks without BN in the convolutional experiments on CIFAR-10. This demonstrates that CWA does provide some of the benefits of normalization. In the fully connected case, the best performer is an affine network with no normalization at all, so that we cannot draw a strong conclusion regarding normalization from the CWA+BN result.
>
> Comment: Essentially, the CWA changes the definition of a layer in a neural net. Do authors see a path from “CWA works” to “CWA works better than affine?”.
>
> Response: Although an interesting question, this has not been our main focus and we see no such path at this time, though there may be one. One possibility is to consider more diverse patterns of connectivity, using the brain as a guide. Inhibitory and excitatory connections are not equally treated in the brain, but appear in specific patterns that we have not replicated in this work. Furthermore, networks in the brain generally have more connections, which may make the CWA normalization less sensitive to random variations in the input. Also, if neuroscientists are going to create more realistic hybrid biological neuronal models that do useful things they will have to incorporate CWA principles to be functioning like real neurons; thus CWA is intended as an example for use by neuroscientists. CWA may also prove useful for physical implementation in neuromorphic chips as well, since it can be reduced to a basic circuit without a need for an ALU.

---

### Decision · Program_Chairs · 2019-12-19

**Decision:**

Reject

**Comment:**

The paper studies non-spiking Hudgkin-Huxley models and shows that under few simplifying assumptions the model can be trained using conventional backpropagation to yield accuracies almost comparable to state-of-the-art neural networks. Overall, the reviewers found the paper well-written, and the idea somewhat interesting, but criticized the experimental evaluation and potential low impact and interest to the community.  While the method itself is sound, the overall assessment of the paper is somewhat below what's expected from papers accepted to ICLR, and I’m thus recommending rejection.